# Addressing food insecurity among U.S. refugees, considering the temporal patterns of food insecurity after resettlement: Qualitative insights from Utah

Nasser Sharareh[1]*, Rachel Dalrymple[2], Konstantinos N. Kambouris[3], Sierra Govett[4], Sarah Adams[4], Jacqueline M. Kent-Marvick[5,6], Rebecca G. Kim[7], Olutobi A. Sanuade[1], Fernando A. Wilson[1,8,9], Andrea S. Wallace[5], Jorie M. Butler[2], Sara E. Simonsen[6]

1 Department of Population Health Sciences, Spencer Fox Eccles School of Medicine at the University of Utah, Salt Lake City, Utah, United States, 2 Department of Biomedical Informatics, Spencer Fox Eccles School of Medicine at the University of Utah, Salt Lake City, Utah, United States, 3 Department of Political Science, College of Social and Behavioral Science, University of Utah, Salt Lake City, Utah, United States, 4 International Rescue Committee (IRC), Salt Lake City, Utah, United States, 5 National Clinician Scholars Program, Perelman School of Medicine, University of Pennsylvania, United States, 6 College of Nursing, University of Utah, Salt Lake City, Utah, United States, 7 Divisions of Gastroenterology, Hepatology, and Nutrition, Department of Internal Medicine, Spencer Fox Eccles School of Medicine at the University of Utah, Salt Lake City, Utah, United States, 8 Department of Economics, College of Social and Behavioral Science, University of Utah, Salt Lake City, Utah, United States, 9 Matheson Center for Health Care Studies, University of Utah, Salt Lake City, Utah, United States.

* nasser.sharareh@hsc.utah.edu

## Abstract

### Background

Refugees experience high rates of food insecurity (**FI**) and its associated health outcomes, such as depression and hypertension. Prior research has identified barriers in accessing food among U.S. refugees. What remains unknown is when accessing food becomes a problem for U.S. refugees and what their preferred strategies are to address FI. Therefore, the objectives were to explore FI experiences among refugees to identify time points at which accessing food becomes a problem and to identify refugees' preferred strategies to address FI.

### Methods

In collaboration with one of the U.S. resettlement agencies in Utah, refugees were recruited for semi-structured interviews using convenience and snowball sampling. Thirty-six interviews were conducted between July and September 2024, in four different languages: English (4 interviews), Dari (6), Arabic (12), and Kinyarwanda (14). Interview transcripts were analyzed using thematic analysis.

**Data availability statement:** Participant characteristics are within the paper(Table 1). De-identified transcripts are available in a public repository: https://figshare.com/articles/data-set/De-identified_English_Transcripts_of_36_interviews/29318549.

**Funding:** We received funding from the University of Utah's School of Medicine: FY24 Vice President for Research Incentive Seed Grant Competition. The funders had no role in study design, data collection and analysis, decision to publish, or preparation of the manuscript.

**Competing interests:** The authors have declared that no competing interests exist.

## Results

FI was at its peak among refugees at four time points. First, when they found their first job in the U.S. Second, after six months in the U.S., when they had to renew their Supplemental Nutrition Assistance Program (**SNAP**) application. Third, when they were no longer receiving caseworkers' support from resettlement agencies. Fourth, when they faced fluctuations in employment or household expenditures. Refugees' preferred strategies to address FI were addressing language barriers, providing a champion to check on them frequently and help when needed, providing information on addressing unmet needs, extending and expanding SNAP benefits, and providing gardens to grow food.

## Conclusion

Four time points when refugees are at higher risk of FI were identified. Community organizations, policymakers, and resettlement agencies should therefore develop interventions to address FI among refugees, specifically around these four time points and informed by refugees' preferred strategies.

## Introduction

Refugees are people who have been compelled to leave their homeland. They cannot go back because they have a legitimate fear of being persecuted for reasons of race, religion, nationality, political opinions, or membership in a particular social group [1]. Since 1975, the United States (U.S.) has accepted refugees [2] and has resettled more refugees than any other country [3]. Over the past four years, nearly 200,000 refugees have entered the U.S. [2]. The State of Utah has been among the top quarter of states in refugee arrivals per capita over the past 10 years [4]. More than 65% of newly arrived refugees in Utah from 2022 to 2024 were from Afghanistan, the Democratic Republic of Congo, Syria, and Venezuela [5], which also reflects the broader trends in refugee resettlement across the U.S. [3,6].

The U.S. refugees are authorized to work in the U.S., can access public benefits, and will receive resettlement services. However, once resettled in the U.S., refugees often experience difficulties in accessing adequate food, a condition defined as food insecurity (**FI**). FI rates among various refugee groups can reach up to 85% [7], which is almost six times the rate observed in the U.S. general population (13.5% in 2023) [8]. Addressing FI among refugees can have a significant public health impact for at least two reasons. First, people with FI may develop unhealthy coping strategies such as excessive use of processed and high-fat foods [9,10] and consumption of fewer healthy foods such as fruits and vegetables [11,12]. These unhealthy diet practices can increase the risk of a variety of chronic diseases such as hypertension, diabetes, and obesity [13–16]. Second, the pressure of stretching a limited budget to put food on the table creates a significant mental burden for adults, which can lead to anxiety and depression [17]. In fact, refugees face a higher prevalence of depression,

hypertension, and obesity one year after arrival compared to the first time they arrived in the U.S. [18], and those with FI have an even higher risk of depression [19]. Therefore, addressing FI among refugees should be a public health priority.

Previous research has indicated that the primary causes of FI among refugees include economic hardship, drastic changes in and lack of information about food choices in the U.S., geographical barriers to accessing grocery stores, lack of or limited employment opportunities, and language barriers [19–25]. However, there is limited understanding of the specific time points at which accessing food becomes a problem for refugees after their arrival in the U.S. This gap in knowledge hinders the ability to design and implement timely interventions to prevent or reduce FI among refugees. Also, less is known about refugees' preferred strategies to address FI. Identifying these priorities can improve the effectiveness of future interventions.

We hypothesized three time points as pivotal moments for FI based on the structure of the U.S. resettlement system. First, upon arrival in the U.S., refugees are welcomed by 10 different resettlement agencies, such as the International Rescue Committee (**IRC**) and Catholic Community Services (**CCS**). Caseworkers in these agencies enroll nearly all the refugees into the Supplemental Nutrition Assistance Program (**SNAP**) [26], which is the largest U.S. nutrition assistance program. Caseworkers also provide training on grocery shopping and using public transportation during refugees' first few months in the U.S. [21]. However, this support usually lasts only up to three months [23], while most refugees may require further support after this time. Second, most SNAP applications must be renewed every 6 months [27] or when there are changes in household conditions, such as income or family size. At this time, many refugees may encounter barriers to renewing their SNAP benefits, as most have not received training on how to apply for SNAP independently. Third, the primary focus of resettlement agencies is securing economic self-sufficiency among refugees [23] by providing job training, revising résumés, and providing mock interviews. However, due to the high demand for these services and the limited availability of caseworkers, both in terms of the short duration of their assistance and the high turnover rate, only some refugees can find employment while they are supported by resettlement agencies' caseworkers. But in many cases, these jobs are undesirable and unstable [28]. Therefore, this new employment results in lowered SNAP benefits, while the income earned from these jobs is not enough to cover all expenses, including food. Moreover, refugees may not have caseworkers' support when they lose their unstable jobs. Therefore, they will face barriers in finding another job and accessing food assistance programs.

In collaboration with the IRC in Salt Lake City, Utah, we explored FI experiences among refugees to identify time points at which accessing food becomes a problem and to identify refugees' preferred strategies to address FI. Results may inform community organizations, policymakers, and resettlement agencies to design timely interventions aimed at reducing FI among refugees.

## Methods

The Institutional Review Board at the University of Utah approved this project (IRB_00176706). Methods for data collection, analysis, and reporting were guided by the Consolidated Criteria for Reporting Qualitative Research (COREQ) guidelines (see S1 Table for details) [29].

### Recruitment

From July 7 to September 19, 2024, we disseminated our recruitment flyer at the IRC and Utah Refugee Center (a nonprofit, state-level organization that provides additional support to refugees in Utah), and through social media. The flyer included information about the duration of the interview, the purpose of the study, a QR code to the screener survey, the telephone numbers of three bilingual interviewers, and that there would be compensation for participation. We also used a snowball sampling approach and used our bilingual interviewers' social network (two of whom were recent refugees) to recruit study participants. The inclusion criteria for the study were: (1) being an adult refugee responsible for meal preparation, grocery shopping, and applying for food assistance programs, and (2) being able to speak Arabic, Dari, English, or Kinyarwanda. These languages were selected based on an input from IRC staff to include refugee communities in Utah

that are both common and likely to experience the most FI problems: Dari for refugees from Afghanistan, English and Kinyarwanda for refugees from the Democratic Republic of Congo and other African countries, and Arabic for refugees from Syria and other Arabic-speaking countries. This approach also ensured that our study included populations that mirrored broader trends in refugee resettlement in the U.S., where recent refugees have predominantly come from Afghanistan, the Democratic Republic of Congo, and Syria [3,6]. Refugees could either fill out a screener survey using the flyer or call our bilingual staff for assistance. We then called the eligible refugees to recruit them and schedule a meeting for a later time with those who were interested in participating in an audio-recorded, semi-structured interview.

## Data collection

We conducted 36 semi-structured interviews with refugees in their preferred language using three bilingual staff members. These staff members were provided with up to 20 hours of training on Human Subject Research and conducting interviews. They were also given guidance on how to translate and transcribe the interviews. Only one interview was conducted at the IRC, while the rest were conducted in the homes of refugees. Most interviews lasted an hour. Refugees were paid $50 for their time. Before the interview, we provided them with a written consent cover letter and refugees provided oral consent, which was recorded and transcribed This procedure was approved by the IRB. At the time of some interviews, the children of participants were in the room as well.

## Instruments

The interview questions included two parts: Part A asked about demographics, FI and nutrition insecurity rates, utilization of food resources, and FI-related health outcomes. After collecting demographic data from participants, we used the Hunger Vital Sign, a validated questionnaire, to assess FI [30]. This questionnaire has two questions: "Within the past 12 months, we worried whether our food would run out before we got money to buy more," and "Within the past 12 months, the food we bought just didn't last and we didn't have money to get more." Response options included often true, sometimes true, or never true. Participants who answered 'often true' or 'sometimes true' to either question were considered to have FI. For nutrition insecurity, we used a 1-item screener developed by the Center for Nutrition & Health Impact [31]: "In the last 12 months, we worried that the food we were able to eat would hurt our health and well-being." Response options included never, rarely, sometimes, often, always, and don't know. If participants answered 'often' or 'always', they were considered to have nutrition insecurity. For access to SNAP, we asked: "At any time in the last 12 months, did you or any family members receive SNAP (food stamp) benefits?" (Yes/No). For emergency food providers' utilization, we asked: "At any time in the last 12 months, did you or any family members ever get free groceries or meals from a food pantry, food bank, church, or other place that helps with free food and meals?" (Yes/No). To assess FI-related health outcomes, including diabetes, hypertension, and heart disease, we used screener questions from the National Health Interview Survey: "Has a doctor ever told you that you had type 2 diabetes or hypertension or coronary heart disease?" (Yes/No). We also asked refugees to rate their English language skills from 1, very good, to 5, very bad.

Part B included open-ended questions to explore FI experiences among refugees to identify time points at which refugees are at higher risk of FI, and to determine refugees' preferred strategies to address FI. For instance, we asked, "Can you tell me when food became a problem?" to explore temporal patterns of FI. We also asked, "What do you think resettlement agencies and other refugee organizations can do to ensure refugees have enough food?" and "How do you think we can ensure that refugees can access food assistance programs when they need help with food?" to seek refugees' preferred strategies to address FI.

## Data analysis

The responses to Part A questions were summarized and reported as percentages. The bilingual staff translated and transcribed Part B of the interview audio recordings into English for analysis. The qualitative analysis team consisted of

six researchers (NS, RB, KK, JMK, JMB, SES; two males and four females) with expertise in FI, community-engaged research, and behavioral science, and all of them were experienced in conducting qualitative analysis. We used thematic analysis to analyze the transcripts in Dedoose (Version 9.2.012). Thematic analysis has six steps [32,33]. In Step 1, we reviewed the transcripts to familiarize ourselves with the data and to identify potential codes and themes. In step 2, we analyzed three transcripts together and created an initial codebook. After the first three transcripts, two coders (RB and KK) coded the remaining transcripts. We then reviewed all the codes and resolved disagreements in code names through discussion and consensus. In step 3, we grouped similar codes together to represent preliminary themes and discussed a potential list of themes. In step 4, we evaluated, modified, and merged themes so that the final themes were meaningful and did not overlap with one another. In this step, we also evaluated the thematic saturation (i.e., the point at which transcripts do not reveal new information) by looking into the codes and themes of the last three transcripts and found that saturation was reached. Therefore, at that point, we ended recruitment. In step 5, we identified final themes for FI experiences, temporal patterns of FI, and refugees' suggested strategies to address FI, with a description of code groupings, sample quotes, and theme descriptions. In step 6, we wrote this manuscript to disseminate the results.

## Results

Fifty-six refugees filled out the screener survey. Of those, seven were ineligible, seven could not be reached, and six were not interested in participating in an audio-recorded interview or were busy at that time. Therefore, we recruited and interviewed 36 refugees.

Table 1 describes participants' characteristics. Among the interviewed refugees, 20 were resettled by IRC, 14 were resettled by CCS, and 2 were from other local organizations that provided refugee resettlement services. Twenty-one of the refugees had been in the U.S. for more than a year, while 15 had been in the U.S. for less than a year. Twelve refugees did the interview in Arabic (nine from Syria), six did it in Dari (all from Afghanistan), four did it in English, and 14 did it in Kinyarwanda (12 from the Democratic Republic of Congo). Refugees were predominantly female (n = 25), insured (n = 30), married (n = 27), in households with an income below 185% of the federal poverty level (n = 30), in households with a car (n = 25), and in households with children under 18 (n = 32).

The proportion of participants experiencing FI was high, with 83% reporting FI experiences in the past 12 months. Also, 22% reported nutrition insecurity. Refugees were well connected to SNAP; 27 of them reported using SNAP in the past 12 months. Three refugees had not used SNAP in the past 12 months, although they had an income below 130% of the federal poverty level (income eligibility criteria for SNAP in Utah). Food pantry utilization was very low, with only 12 refugees using pantries in the past 12 months. Of those 24 refugees who did not use food pantries, 19 had an income below 185% of the federal poverty level. Also, only 12 refugees participated in the IRC's New Roots Program [34], a program that assists refugees to become farmers and grow cultural foods and later sell their products in Farmers Markets. The remaining 24 had not heard of New Roots or had not participated in it. Refugees also reported diet-related diseases: 14% had diabetes, 17% had hypertension, and 3% had heart disease.

Through part B of the interviews, we explored FI experiences among refugees to identify time points at which refugees are at higher risk of FI and sought refugees' preferred strategies to reduce their FI. For each theme, we present two quotes followed by participant ID, language, and sex. See S2 Table, which includes additional quotes related to each theme.

### Themes relating to FI experiences

Six themes emerged from FI experiences. These themes were found to map closely to an established theoretical framework of access to services [35,36], and therefore, we adopted their terminology. The six themes are: Affordability, Awareness, Accommodation, Acceptability, Availability, and Accessibility.

**Table 1. Population characteristics of the interviewed refugees (N = 36).**

| Variable | % (N) |
|---|---|
| **Resettlement agency** | |
| **International Rescue Committee (IRC)** | 55% (20) |
| **Catholic Community Services (CCS)** | 39% (14) |
| **Others** | 5.5% (2) |
| **Language** | |
| **Arabic** | 33% (12) |
| **Dari** | 17% (6) |
| **English** | 11% (4) |
| **Kinyarwanda** | 39% (14) |
| **Average English language skills** | 3.8 (1 is very good; 5 is very bad) |
| **County of origin** | |
| **Democratic of Congo** | 33% (12) |
| **Afghanistan** | 19% (7) |
| **Syria** | 25% (9) |
| **Others (Burundi, Central Africa Republic, Iraq, Sudan, Uganda)** | 22% (8) |
| **Length of U.S. residency** | |
| **Less than a year** | 42% (15) |
| **More than a year** | 58% (21) |
| **Average age** | 37 (Min: 16 – Max: 57) |
| **Sex** | |
| **Female** | 69% (25) |
| **Male** | 30% (11) |
| **Married** | 75% (27) |
| **Race** | |
| **Black or African American** | 47% (17) |
| **Asian** | 19% (7) |
| **Middle Eastern** | 33% (12) |
| **Education** | |
| **Less than high school** | 33% (12) |
| **High school degree** | 25% (9) |
| **Some college or above** | 25% (9) |
| **Missing** | 17% (6) |
| **Employed (part-time or full-time)** | 42% (15) |
| **Average household size** | Average: 5 (Min 1 – Max 10) |
| **Children in household** | 92% (33) |
| **Children in household under 18** | 89% (32) |
| **Children in household under 5** | 44% (16) |
| **Income less than 185% of the federal poverty level** | 83% (30) |
| **Income less than 130% of the federal poverty level** | 75% (27) |
| **Sending remittance** | 44% (16) |
| **Insured** | 83% (30) |
| **Had a disability** | 11% (4) |
| **Access to a private vehicle** | 69% (25) |

*(Continued)*

**Table 1.** (Continued)

| Variable | % (N) |
|---|---|
| **Food insecurity in the past 12 months** | 83% (30) |
| **Nutrition insecurity in the past 12 months** | 22% (8) |
| **SNAP utilization in the past 12 months** | 75% (27) |
| **Food pantry and other free food resources (e.g., church) utilization in the past 12 months** | 33% (12) |
| **New Roots participation** | 33% (12) |
| **Diabetes** | 14% (5) |
| **Hypertension** | 17% (6) |
| **Heart disease** | 3% (1) |

**Affordability (an income-related barrier):** This theme refers to refugees' perception of the worth of a service relative to cost and their ability to pay for the service. The theme of Affordability reflects income-related barriers, including the cost of food, the inability to afford food, unemployment due to limited opportunities or language barriers, and ineligibility for federal nutrition assistance programs such as SNAP for individuals whose income slightly exceeds program thresholds.

*I was trying to find a decent job that offered good pay. Both my daughter and my wife have applied at places like company A, B, and C (we removed the company names to ensure confidentiality), which are all part of a referral service that helps you get registered and find jobs. We went there hoping they could help us find work, but they were unable to assist us. The main problem was our lack of language skills, which made it very difficult. If we didn't have a language barrier, perhaps we would have been accepted in some position (#18, Dari, Male).*

*They cut off the food stamps and we are left with nothing. It was the only source of comfort for me. They cut off the food stamps because we have a higher income now (#85, Arabic, Female).*

**Awareness (a non-income-related barrier)**: This theme refers to informational barriers such as limited knowledge about the U.S. food environment and food programs, including their eligibility criteria and application process.

*I didn't apply for food stamps because we thought we might not qualify. It felt overwhelming, and we were unsure about the process, so we decided to hold off on applying (#19, Dari, Female).*

*What prohibits them [other refugees], like my family members, is not knowing about those programs. If for example, you don't tell me about something, I can't know about it. But if someone tells me about it, then I will know about and use it if I need it (#44, Kinyarwanda, Female).*

Also, many refugees were not using food pantries simply because they did not know how to access them.

*There is a lot of stuff that you mentioned that I just did not know about. IRC only told us about food stamps, but they didn't tell us about food pantries or churches like you mentioned. (#83, Arabic, Male)*

*I did not know about them [food pantries]. That's what I told you; they are some information we don't get to know. (#38, Kinyarwanda, Male)*

**Accommodation (a non-income-related barrier)**: This theme refers to whether the providers can meet refugees' needs and can include translation services, technology, or the provision of cultural foods.

*We told them [resettlement agency] that we needed help in learning English. They told us they did not bring us here to teach us, we had to work and learn from experience. We were in shock. We had no idea where we were going. They gave us a bus pass and taught us how to use maps, but it was very hard (#82, Arabic, Female).*

*It is very difficult because several people are uneducated. If they go to the IRC or Department of Workforce Services (DWS), they might not be able to communicate properly and may struggle with the translation when applying (#18, Dari, Male).*

One of the advantages of living in Utah was the provision of cultural foods (foods that meet the tastes and needs of a particular culture) in grocery stores and Farmers Markets. Specifically, refugees started accessing cultural foods when they lived longer in Utah and became familiar with resources. However, some refugees were not using pantries because they did not have cultural foods.

*I get my cultural foods like vegetables and fresh beans because you know us Africans, we like fresh food. So, things like eggplants, spinach, and other vegetables, I get them there [Farmers Markets] (#88, Arabic, Female).*

*Initially, when we arrived, we didn't eat any outside food. Over time we discovered that some of those familiar foods can be found here as well, and we have started to use them. Gradually things are getting better as we learn more about the local culture and foods. We also discovered halal markets here. At first, we thought there was no such thing as fresh bread, but we learned that they bake it in the halal markets. This experience has opened our eyes to the variety of options available that we didn't realize existed at first (#19, Dari, Female).*

**Acceptability (a non-income-related barrier)**: This theme refers to refugees' attitudes and comfort with the services and service providers. It can include discrimination and stigma. Refugees did not report any problems that could indicate they were discriminated against. However, stigma existed among some of the refugees, whether due to their limited English language skills, wearing a hijab or a different cultural costume, or being shy to ask for help.

*I used to be scared to go out, especially since I was a hijabi and someone living in a culture that isn't their own. My caseworker used to encourage me to go out and not be afraid (#91, Arabic, Female).*

*I honestly think that they don't tell me to tell them when I need food that I should talk to them. They should tell us that they are able to give us food assistance when we need it. They should have told us not to be shy and that this was our right. There are days when people go to IRC and tell them they don't have food. Some people try and wait it out until they get food stamps again (#94, Arabic, Female).*

**Availability (a non-income-related barrier)**: This theme refers to sufficient services and resources to meet refugees' needs, including availability of food resources in their neighborhood (e.g., Farmers Markets) and staff availability and capacity to assist refugees with their unmet food needs. In Utah, one of the resources that was instrumental in connecting refugees to SNAP, specifically when they were no longer supported by caseworkers at resettlement agencies, was the SNAP office at the Department of Workforce Services (DWS).

*Our caseworker gets changed every three months. As soon as our situation gets settled by the caseworker, they change them (#81, Arabic, Female).*

*DWS is the one that helped me. They are the ones who helped me when I reached here, so when I have problems, I go to them so they can help me. There's no one else to help me except them. They are the ones who request things from the bank, check stubs, things related to work, and many others. They requested everything, including rent payments,*

*how much we pay, and how much we make. When they saw that the assistance was low, they opened those things (#44, Kinyarwanda, Female).*

**Accessibility (a non-income-related barrier)**: This theme refers to whether refugees can access food. Barriers and facilitators may include travel time to food resources, access to a vehicle or public transportation, cooking appliances, and cooking skills.

*Honestly, with a car, they are close, but without a car, I have to take the bus, and that is far. I go to company D and C. I got used to company D and C. It takes me two buses to get to company D. The waiting time is what takes the most time when taking the bus (#80, Arabic, Male).*

*In the heart of the winter, I would walk to Company C, and we would get home wet and cold after getting groceries. Sometimes I talk to the kids about the times we used to go to company E walking. I used to take my son on his bicycle and put the groceries on the bicycle and make him go ahead of me (#91, Arabic, Female).*

### Themes on temporal patterns of FI

After identifying themes on FI experiences, we looked for time points when FI was at its peak among refugees. Four different themes emerged: finding the first job, SNAP application renewal, losing caseworkers' support, and employment fluctuations and household expenditures. Table 2 shows the underlying FI problem of these time points and the strategies refugees suggested to address those problems.

Table 2 is based on authors' analysis of the transcripts. Refugees also suggested other strategies such as providing free bus passes, training for using technology and computers, and job opportunities other than farming. However, these strategies were not mentioned frequently enough to be considered a major theme.

**Finding the first job**: As soon as refugees had found a job, they lost all or some part of their SNAP benefits, even though they were still unable to afford their household expenses. The theme directly refers to the Affordability issues (see Table 2).

**Table 2. Time points at which refugees are at higher risk of food insecurity, their underlying food insecurity problem, and refugees' preferred strategies to address those problems.**

| Time Point | Underlying Food Insecurity Problem | Refugees' Preferred Strategy |
|---|---|---|
| **Finding the first job** | Affordability: Income-related barriers | Extending or expanding SNAP benefits<br>Providing gardens to grow food |
| **SNAP application renewal** | Availability: Availability of staff to assist refugees<br>Awareness: Informational barriers | Addressing language barriers<br>Providing a champion<br>Providing information on how to address their unmet needs |
| **Losing caseworkers' support** | Accommodation: Refugees' preferences such as translational services<br>Availability: Availability of staff to assist refugees | Addressing language barriers<br>Providing a champion |
| **Employment fluctuations and household expenditures** | Affordability: Income-related barriers<br>Availability: Availability of staff to assist refugees<br>Awareness: Informational barriers | Providing gardens to grow food<br>Providing a champion |

*When my husband started working and they started lessening our food stamp amount, that's when we broke. We were not able to keep up with the demand of food in our household. We could not get enough food for the month. We tried to adjust to the new situation, but it would not work out for us (#81, Arabic, Female).*

*My husband works at the moment and I would love to go work part-time to improve my language. But then I think about it, I will not be able to work a job that gets me a high paycheck, but at the same time, I will harm myself. Food stamps will be cut off. So, whatever I make from my hard work will be used for compensating the lack of food stamps. I will have to take into account my kids, my house, and other financial costs. So, I always am unsure if I want to work or not. I don't know if my point of view is different or if someone may look at it differently. But that is how I feel (#90, Arabic, Female).*

**SNAP application renewal**: Most SNAP applications must be renewed every 6 months or every time there is a change in household conditions such as income or family size. However, refugees are not familiar with these requirements and often fail to submit their application correctly or in time. This theme refers to the Awareness, Availability, and Accommodation issues (see Table 2).

*I just woke up one day and found when they had closed everything. They told me that after every 6 months, people must go and reapply for food stamp (#40, Kinyarwanda, Male).*

*Our food stamps have been cut off, we went to the refugee center and called to get it sorted out. It took almost a month, about 20 days, to get it fixed. This was very stressful. We didn't have any other worries. Two or three months ago when we moved some documents were lost, and we should have reported within six months. Because we didn't, our food stamps were cut off. Then we went back, filled out the forms, and finally it was resolved (#17, Dari, Female).*

**Losing caseworkers' support**: Refugees rely on caseworkers' support to access resources. However, high turnover among caseworkers and the limited time they have to assist refugees (both in terms of high demands and the time period during which caseworkers' support is available) can reduce refugees' access to resources. This theme refers to the Availability and Accommodation issues (see Table 2).

*Our caseworker gets changed every three months. As soon as our almost situation gets settled by the caseworker, they change them (#81, Arabic, Female).*

*I'm talking in general, not just IRC. They don't help us like the way we are told when we are still in Africa. Because after staying here for a specific time, they just leave you. Then you go under the care of Refugee Center, but they also have other things they are responsible for (#38, Kinyarwanda, Male).*

**Employment fluctuations and household expenditures**: Refugees do not always find stable jobs and may face unemployment frequently. Due to limited access to their caseworkers, specifically when they are no longer supported by resettlement agencies, securing employment after a job loss is a significant barrier. In addition, expected and unexpected costs, like those from healthcare, can limit their budget for food. This theme refers to the Affordability issues and can be exacerbated by the Availability and Awareness issues (see Table 2).

*There are times when you face challenges of sickness and then you have to pay medical bills in addition to buying food. For example, I had challenges of sickness, and now I have to pay a lot of medical bills, which makes it hard to get enough money for groceries (#30, Kinyarwanda, Female).*

*The problems started when my job ended around March of this year. When I was working, we could put money together and pay for everything, buy what we want to eat. Food became a problem because now that it's only one person working (#43, Kinyarwanda, Female).*

**Themes on suggested strategies by refugees**

Finally, we asked refugees what they think is needed to reduce/address their FI. Five strategies emerged:

**Addressing language barriers**: Language barriers among refugees not only impact their employment but also hinder their access to food benefits and resources. This strategy refers to addressing Accommodation issues, which can eventually address Affordability and Awareness issues (see Table 2).

*The main problem was our lack of language skills, which made it very difficult. If we didn't have a language barrier, perhaps we would have been accepted in some position (#18, Dari, Male).*

*We have people in Syria that didn't go to college, and they are super poor. They would learn 3 or 4 technical jobs. They would learn how to clean sewers or construction or farmer. Even the lowest people in Syria know how to farm. Everyone knows how to work. Then use their experience here and help them get a job instead of them staying at home. But no, instead this job has a language barrier, this job wouldn't work for us. They destroyed us mentally. Everyone who had a big drive within them was met with a giant wall that just stopped them. On the inside they destroyed us mentally because we came here already with a destroyed mental state (#85, Arabic, Female).*

**Providing a champion (someone who supports a cause or person) to check on them frequently and help when needed:** A second preferred strategy was having a champion or someone who could frequently check on refugees to ensure they have enough food, are connected to resources, and can address other challenges. This theme refers to addressing Awareness, Availability, and Acceptability issues (see Table 2).

*It would be great if someone like you could visit each family to ask about their problems. This way, they can share their concern if they have any. If not, it reassures them that everything is fine, and they feel safe without the burden of psychological stress. Food insecurity is a significant difficulty, I am glad you came to our house and allowed me to express my problems (#15, Dari, Male).*

*They have to ask refugees every month and check if they are in need of food or something. Refugees don't have it in mind to ask because they are somewhere new, they don't know anything, they are lost. They have to be able to check up on them. As long as I have kids I am in need of food (#94, Arabic, Female).*

**Providing information on how to address their unmet needs**: Many refugees mentioned they do not know where and how they can get assistance with food and other needs. Some mentioned they do not even know how to apply for food programs like SNAP. This theme refers to addressing Awareness issues (see Table 2).

*Informing us of places that can help. After that they can get help from places. First off, there is a language barrier. There aren't people to tell us where you can get help, gaining information is not accessible for those who need it (#83, Arabic, Male).*

*I would like to learn how to apply for benefits like food stamps, Medicaid, and other things to help me in life (#32, Kinyarwanda, Female).*

**Providing gardens to grow food**: Considering a strong background in farming among refugees, some asked for gardens to grow their food. This theme refers to addressing Affordability and Accommodation issues (see Table 2).

*I think what can improve food security in refugees is farming. Given the fact that life is hard here, if they can give us like small gardens, so we can grow our vegetables, put food in storage and use it when needed, those things can help us a lot (#32, Kinyarwanda, Female).*

*Giving people small gardens to do some farming like vegetables like carrots, tomatoes, cabbage, would be great (#38, Kinyarwanda, Male).*

**Extending and expanding SNAP benefits**: Not surprisingly, many refugees asked for extended duration for SNAP benefits after they arrive in the U.S. They also asked for more benefits and expansion of income eligibility criteria for SNAP. This theme refers to addressing the Affordability issues (see Table 2).

*If you are asking for my opinion, then my opinion is that food stamps are an essential thing that everyone needs. I wish that they would change the fact that even if someone started working, they would still have access to them. Because this work shouldn't just be for basic needs. You are saving money, you are building your future, you are helping your kids, helping your husband, maybe helping people who lived in another country (#90, Arabic, Female).*

*The only thing is that they cut off people as soon as they start to have an income. I wished that we still had them, especially since we are new. They cut off the food stamps and we are left with nothing. It was the only source of comfort for me. They cut off the food stamps because we have a higher income now. I told the caseworker, I told her that yes, our income is higher but rent is high, we had to get a car because of how bad the transportation was (#85, Arabic, Female).*

## Discussion

While prior research has identified barriers to food access among different refugee populations in the U.S. [19–25], our major contributions in this paper are identifying the time points at which accessing food becomes a problem for refugees after their arrival in the U.S., and seeking refugees' preferred strategies to address FI.

We found that both income- and non-income-related barriers contributed to refugees' FI problems. Income-related barriers (i.e., Affordability) were the main causes of FI, including the cost of food, unemployment, and ineligibility for SNAP. Non-income-related barriers, including Accommodation, Availability, and Awareness, exacerbated FI problems as they limited access to food resources designed to address FI, such as SNAP and food pantries. Accommodation, Availability, and Awareness issues mainly reflected informational and language barriers among refugees, which became more noticeable when resettlement agencies' caseworkers no longer supported refugees.

Prior research over the past two decades has identified the same income- and non-income-related barriers [19–25]. One of the main reasons for the persistence of these barriers is probably the limited support available to resettlement agencies. For instance, in Utah, resettlement agencies and organizations have caring, dedicated, and well-trained staff who know how to address the needs of refugees. However, resettlement agencies have been historically underfunded across the country [28,37,38], regardless of the political party in power, limiting their ability to address refugees' needs. The impact of these barriers has become even more significant in 2025 and could become worse in the future because the current U.S. administration has suspended refugees' admission to the U.S. starting January 27, 2025 [39]. Consequently, caseworkers' support has become very limited, as a large portion of resettlement agencies' funding comes from the Office of Refugee Resettlement based on the number of newly admitted refugees [23]. Also, much of the funding for refugees who are already in the U.S. has been terminated as well.

The U.S. resettlement system has been suspended to evaluate whether it would be in the interests of the U.S. However, according to a recent report by the U.S. Department of Health and Human Services, the net fiscal impact of refugees and asylees from 2005 to 2019 has been $123.8 billion [40]. It means that refugees and asylees contributed more revenue than they cost in expenditure to the government. This massive contribution suggests that we can still improve our current resettlement system, without compromising the availability of resources to Americans, by allocating more funding to resettlement agencies so that they can provide the support refugees need to thrive and integrate faster into the American society.

We also found four time points at which refugees are at higher risk of FI. The underlying FI problems among three of those four time points (losing caseworkers' support, employment fluctuations, and SNAP renewal) were non-income-related barriers, including Accommodation, Availability, and Awareness. This finding further highlights the need for additional funding for resettlement agencies to focus on language barriers and provide extended support and more information to refugees. Also, the strategies refugees suggested were, not surprisingly, focused on addressing these barriers.

For instance, many refugees requested addressing their language barriers as it will both improve their employment opportunities [41] to prevent FI and increase their access to resources if they experience FI. Resettlement agencies help refugees to apply for classes in English as a Second Language (ESL). However, classes are mostly provided on weekdays, when refugees cannot take time off from their work, or they are offered in the evening, when refugees may be tired and sometimes hungry, thus they cannot focus on learning a new language. Also, most of their time during the first few years in the U.S. is focused on finding and maintaining employment, which prevents them from attending in-person classes [42], while research has shown that the most improvement in learning a new language occurs during the early years of resettlement [43]. In addition, these classes fail to meet the demands of the labor market as they focus on generic language skills [44]. With recent advancements in artificial intelligence and Mobile-Assisted Language Training, the U.S. resettlement system should adopt new approaches to address language barriers by offering online and work-focused English language classes. New and effective approaches to addressing language barriers could potentially have the most significant impact not only on reducing FI but also on achieving economic self-sufficiency, which has been the focus of the U.S. resettlement system.

Refugees also requested a champion. Having a champion, whether from their community or hired through the resettlement agencies, can make a huge difference because many refugees were not seeking help due to their pride, stigma, or travel time to resettlement agencies. Additionally, some participants had limited English language skills to seek information on their own. There are many non-profit organizations in Utah, and similarly across the U.S., either for specific communities (e.g., Afghan, Iraqi, Somali) or for the general refugee population (e.g., Utah Refugee Center, Utah Refugee Connection) that provide services to refugees. All these organizations have a very limited budget. Empowering these organizations not only will reduce the burden on resettlement agencies but will also empower those communities to further support refugees.

Additionally, refugees requested more information about resources. Considering the limited availability of caseworkers, not every refugee receives enough support. For instance, the majority of our participants were not using food pantries due to the limited knowledge about their existence or location. Not using food pantries was observed among Afghan refugees in a previous study as well [21]. In most cases, they might receive information in English, while they have limited English language skills. Therefore, providing information, specifically translated information, might ensure that refugees find assistance when needed. Also, some of the refugees faced informational barriers in accessing SNAP (i.e., they had limited information about how to apply for SNAP or where to get assistance on the application.) Developing implementation strategies to address informational barriers, considering the implementation barriers and facilitators that exist in resettlement agencies, could be a promising next step.

Another time point at which refugees are at higher risk of FI is when they find their first job. This time point reflects Affordability issues. Accordingly, refugees requested extended or expanded SNAP benefits. In Utah, the income eligibility

criterion for SNAP is 130% of the federal poverty level (as of April 2025), while many states use 200% as the threshold. Also, as soon as refugees find their first job, they lose all or some portion of their SNAP benefits while they have not yet reached a stable income and may have only temporary employment. In fact, refugees experience employment declines the longer they stay in the U.S. [28]. Furthermore, many refugees do not have enough savings or a strong social network to help them in difficult times. The inadequacy of SNAP benefits has been well documented in the literature and not just for refugees but for all Americans [45,46]. Therefore, extending or expanding SNAP benefits could significantly and positively contribute to economic self-sufficiency among refugees and reduce the negative health impact of FI.

Refugees also requested gardens to grow their food. This strategy addresses the Affordability issues by providing low-cost, cultural foods to refugees. It can also create an additional source of income, potentially mitigating temporal FI problems caused by employment fluctuations. Such opportunities may be currently available. For instance, IRC pioneered the New Roots program, which exists in 11 U.S. cities as of April 2025 [34]. The New Roots program assists refugees in accessing agricultural land and fresh, local food through community gardens. It also offers a farm business training program, program-facilitated farmers markets, and a Community Supported Agriculture Program. These initiatives provide farmers with low-barrier sales opportunities and supply the local community with fresh, culturally familiar vegetables. Additionally, the program includes a youth agriculture initiative. Yet, only 12 of the participants in this study had taken part in this program. Therefore, future efforts should focus on expanding the capacity of New Roots, connecting more refugees to this program, and providing other farming opportunities to refugees.

Taking all these observations into account, we suggest that the U.S. resettlement system adjust in three ways. First, there must be additional support provided to resettlement agencies and refugee communities to enable them to offer structured and extended support to refugees. Considering the economic contribution of refugees to the U.S. government [40], we believe this approach will be cost-effective. Second, the goal of the U.S. resettlement system is to achieve economic self-sufficiency among refugees within the first three months in the U.S. by finding them a job. This is an unrealistic expectation [23] and has not yielded positive outcomes. Empowering refugees, however, by addressing their language barriers and providing extended caseworkers' support could ultimately yield a higher and stable income for refugees. And third, resettlement agencies should adopt new strategies to address language and informational barriers among refugees. Future research, in collaboration with resettlement agencies, should adopt user-centered design approaches to develop effective implementation strategies to address these barriers, as research has shown that our current approaches, while necessary, are insufficient.

## Limitations

This study has a few limitations. First, some of the codes identified during our analysis could be included in different themes. For instance, cultural food preferences could be considered in either accommodation (whether providers can meet refugees' needs and constraints) or acceptability (refugees' comfort with the resources). Also, cooking appliances could be considered in either accessibility (whether refugees have the resources to receive the service) or availability (availability of resources to meet refugees' needs). With the authors' consensus, we decided which themes these codes belonged to. Second, although we cannot speak to the generalizability of our findings, prior studies have identified similar FI experiences among different refugee populations in the U.S. While we might have reached a vulnerable population due to our recruitment methods (i.e., identifying the communities that we hypothesized could have more FI problems and snowball sampling), we interviewed refugees from Afghanistan, Democratic Republic of Congo, and Syria, to reflect the broader trends in refugee resettlement across the U.S. [3]. Future research should also include refugee communities that might face fewer barriers due to the availability of social support to identify facilitators in accessing food. And finally, although 83% of our sample had experienced FI, we do not think this indicates an 83% FI rate among all Utah refugees. If we had collected more broadly representative data on Utah refugees, this rate would likely have been lower.

## Conclusion

In this paper, we explored FI experiences among refugees to identify time points at which refugees are at higher risk of FI and determined refugees' preferred strategies to address FI. Community organizations, policymakers, and resettlement agencies should develop interventions to address FI among refugees informed by the time points at which refugees are at higher risk of FI and by refugees' preferred strategies for reducing that risk.

## Supporting information

**S1 Table.  The 32-item checklist based on the Consolidated Criteria for Reporting Qualitative Research (COREQ).**
(DOCX)

**S2 Table.  Additional quotes related to themes from the main text.**
(DOCX)

## Acknowledgments

Interviewees: We truly appreciate our part-time staff who conducted the interviews. Thank you, Lin Alsubhi, Juliet Munyambanza, and Mahsa Qaderi.

## Author contributions

**Conceptualization:** Nasser Sharareh, Sierra Govett, Sarah Adams, Fernando A. Wilson, Sara E. Simonsen.

**Data curation:** Nasser Sharareh, Rachel Dalrymple, Konstantinos N. Kambouris.

**Formal analysis:** Nasser Sharareh, Rachel Dalrymple, Konstantinos N. Kambouris, Jacqueline M. Kent-Marvick, Sara E. Simonsen.

**Funding acquisition:** Nasser Sharareh, Sierra Govett, Sarah Adams, Sara E. Simonsen.

**Investigation:** Nasser Sharareh, Sara E. Simonsen.

**Methodology:** Nasser Sharareh, Rachel Dalrymple, Jorie M. Butler, Sara E. Simonsen.

**Project administration:** Nasser Sharareh.

**Resources:** Nasser Sharareh.

**Software:** Nasser Sharareh, Jorie M. Butler.

**Supervision:** Fernando A. Wilson, Andrea S. Wallace.

**Validation:** Nasser Sharareh, Jorie M. Butler, Sara E. Simonsen.

**Writing – original draft:** Nasser Sharareh, Rachel Dalrymple, Konstantinos N. Kambouris.

**Writing – review & editing:** Nasser Sharareh, Rachel Dalrymple, Konstantinos N. Kambouris, Sierra Govett, Sarah Adams, Jacqueline M. Kent-Marvick, Rebecca G. Kim, Olutobi A. Sanuade, Fernando A. Wilson, Andrea S. Wallace, Jorie M. Butler, Sara E. Simonsen.

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
