## [Decision Letter · Decision Letter 0]

PONE-D-25-21762Addressing food insecurity among U.S. refugees: Insights from UtahPLOS ONE

Dear Sharareh,

Thank you for submitting your manuscript to PLOS ONE. After careful consideration, we feel that it has merit but does not fully meet PLOS ONE’s publication criteria as it currently stands. Therefore, we invite you to submit a revised version of the manuscript that addresses the points raised during the review process.

We look forward to receiving your revised manuscript.

Kind regards,

Md. Feroz Kabir, BPT, MPT, MPH, BPED, MPED

Academic Editor

PLOS ONE

Journal Requirements:

2.  In the ethics statement in the Methods, you have specified that oral consent was obtained. Please provide additional details regarding how this consent was documented and witnessed, and state whether this was approved by the IRB

“We received funding from the University of Utah’s School of Medicine: FY24 Vice President for Research Incentive Seed Grant Competition”

“Funding: We received funding from the University of Utah’s School of Medicine: FY24 Vice President for Research Incentive Seed Grant Competition. Interviewees: We truly appreciate our part-time staff who conducted the interviews. Thank you, Lin Alsubhi, Juliet Munyambanza, and Mahsa Qaderi.”

“We received funding from the University of Utah’s School of Medicine: FY24 Vice President for Research Incentive Seed Grant Competition”

5. In the online submission form, you indicated that “De-identified transcripts can be shared upon signing an agreement prior to release.”

**Additional Editor Comments:**

Please submit your revised manuscript within next 15 working days.

Reviewers' comments:

Reviewer's Responses to Questions

**Comments to the Author**

1. Is the manuscript technically sound, and do the data support the conclusions?

Reviewer #1: Yes

Reviewer #2: Yes

2. Has the statistical analysis been performed appropriately and rigorously? 

Reviewer #1: Yes

Reviewer #2: Yes

3. Have the authors made all data underlying the findings in their manuscript fully available?

Reviewer #1: Yes

Reviewer #2: Yes

4. Is the manuscript presented in an intelligible fashion and written in standard English?

Reviewer #1: Yes

Reviewer #2: Yes

5. Review Comments to the Author

Reviewer #1: Comments to Authors

Congratulations on completing this important project, which sheds light on the food security status of refugees and addresses a key gap in existing knowledge. I would be honored to offer a few suggestions that may help strengthen the manuscript, should you find them relevant.

Title

The current title may give the impression that this is a review article, when in fact it reports original research. To better reflect the nature of the study, it is recommended to specify the study design—for example, by indicating that qualitative methods were used. This will help clarify the article's methodological approach and contribution for potential readers. Additionally, the title could be more focused by incorporating key components of the findings, such as the circumstances under which refugees are most vulnerable and their preferred strategies for addressing food insecurity.

Abstract

[Line 26-27] The clear identification of the research gap is a strong aspect of the study. Clearly stating the objective of the study would enhance the clarity and focus of the manuscript.

[Line 28] Including the sampling method would be a valuable addition. It would also be helpful to briefly describe the process of refugee recruitment.

Introduction

[Additional suggestion] The content is well-structured and informative. However, providing additional information on what refugees are permitted or restricted from doing upon arrival in the U.S. would help enlighten international readers and offer a more comprehensive understanding of their resettlement experience.

Methods

[line 105] How about the duration of arriving at U.S.? No exclusion? How about if the refugee come alone? Refugee from food insecure household?

[line 116] Any permission needed for conducting the study at the resettlement centers? The information might be beneficial the future studies which are planned to conduct data collection at the resettlement centers.

[line 116] Could you clarify how the study determined that the number of interviews conducted was sufficient to capture comprehensive and meaningful insights?

[line 120] It is recommended to insert a subheading titled 'Instruments' to clearly present the tools or materials used for data collection.

[Additional suggestion] It would be helpful to include any steps taken to ensure the quality and trustworthiness of the qualitative data. Additionally, providing information about the interviewers—such as their training and experience in conducting qualitative research—would strengthen the credibility of the study.

Results

[line 195] Are there any participant quotes related to 'cost of food' and 'the inability to afford food' provided in the supplementary file? These quotes are not presented in the main text.

[line 200 & 274] It is recommended to remove the company names to protect privacy and ensure confidentiality for the organizations involved.

[line 358] It would be helpful to provide an explanation of the term 'champion,' especially since it may not be familiar to international readers.

Discussion

[line 409] It may be beneficial to categorize the themes in the results section into 'income-related' and 'non-income-related' barriers. This would provide a clearer reflection of the distinct challenges faced by refugees.

[Additional suggestion] How have other countries addressed similar issues? Are there any solutions or best practices from other countries that could be applied or learned from in addressing these challenges?

Conclusion

[Additional suggestion] What is the potential feasibility of implementing the suggested methods in addressing food insecurity (FI)?

Reviewer #2: The methodology should be more clearly stated. The result should be presented with some graphs, not only tables. The strength and limitation should be clearly stated. All through English, grammar correction is needed.

6. PLOS authors have the option to publish the peer review history of their article (what does this mean?). If published, this will include your full peer review and any attached files.

Reviewer #1: No

Reviewer #2: **Yes: **Mohammad Mohinul Islam

---

## [Author Response · Author response to Decision Letter 1]

13 Jun 2025

Response to Reviewers letter is attached.

---

## [Editor Report · Decision Letter 1]

Addressing food insecurity among U.S. refugees, considering the temporal patterns of food insecurity after resettlement: Qualitative insights from Utah.

PONE-D-25-21762R1

Dear Dr. Sharareh,

We’re pleased to inform you that your manuscript has been judged scientifically suitable for publication and will be formally accepted for publication once it meets all outstanding technical requirements.

Kind regards,

Md. Feroz Kabir, BPT, MPT, MPH, BPED, MPED

Academic Editor

PLOS ONE
---

## [Editor Report · Acceptance letter]

PONE-D-25-21762R1

PLOS ONE

Dear Dr. Sharareh,

I'm pleased to inform you that your manuscript has been deemed suitable for publication in PLOS ONE. Congratulations! Your manuscript is now being handed over to our production team.

Kind regards,

on behalf of

Dr. Md. Feroz Kabir

Academic Editor

PLOS ONE